

# Rephine.r: a pipeline for correcting gene calls and clusters to improve phage pangenomes and phylogenies

Jason W. Shapiro[1] and Catherine Putonti[1,2,3]

[1] Department of Biology, Loyola University Chicago, Chicago, IL, United States of America
[2] Department of Microbiology and Immunology, Stritch School of Medicine, Loyola University Chicago, Maywood, IL, United States of America
[3] Bioinformatics Program, Loyola University Chicago, Chicago, IL, United States of America

Corresponding author
Jason W. Shapiro, jshapiro2@luc.edu

## ABSTRACT

**Background**. A pangenome is the collection of all genes found in a set of related genomes. For microbes, these genomes are often different strains of the same species, and the pangenome offers a means to compare gene content variation with differences in phenotypes, ecology, and phylogenetic relatedness. Though most frequently applied to bacteria, there is growing interest in adapting pangenome analysis to bacteriophages. However, working with phage genomes presents new challenges. First, most phage families are under-sampled, and homologous genes in related viruses can be difficult to identify. Second, homing endonucleases and intron-like sequences may be present, resulting in fragmented gene calls. Each of these issues can reduce the accuracy of standard pangenome analysis tools.

**Methods**. We developed an R pipeline called Rephine.r that takes as input the gene clusters produced by an initial pangenomics workflow. Rephine.r then proceeds in two primary steps. First, it identifies three common causes of fragmented gene calls: (1) indels creating early stop codons and new start codons; (2) interruption by a selfish genetic element; and (3) splitting at the ends of the reported genome. Fragmented genes are then fused to create new sequence alignments. In tandem, Rephine.r searches for distant homologs separated into different gene families using Hidden Markov Models. Significant hits are used to merge families into larger clusters. A final round of fragment identification is then run, and results may be used to infer single-copy core genomes and phylogenetic trees.

**Results**. We applied Rephine.r to three well-studied phage groups: the Tevenvirinae (e.g., T4), the Studiervirinae (e.g., T7), and the Pbunaviruses (e.g., PB1). In each case, Rephine.r recovered additional members of the single-copy core genome and increased the overall bootstrap support of the phylogeny. The Rephine.r pipeline is provided through GitHub (https://www.github.com/coevoeco/Rephine.r) as a single script for automated analysis and with utility functions to assist in building single-copy core genomes and predicting the sources of fragmented genes.

## INTRODUCTION

A pangenome is the collection of all genes found in a set of related genomes (*Tettelin et al., 2005*; *Vernikos et al., 2015*). These genomes might be different strains of the same species or taken from the same genus or higher taxonomic level. Pangenomes are useful, because they allow one to compare gene content variation to differences in phenotypes, ecology, and evolutionary history. For instance, by mapping gene content of potential pathogens onto a phylogeny and contrasting clade-specific genes with differences in reported strain virulence, the pangenome can help reveal how these genes relate to pathogenicity while placing them in an evolutionary context (*e.g.*, *Hurtado et al., 2018*; *Wyres et al., 2019*). Pangenomes have also been used to describe which functions are conserved among members of bacterial taxa in different environments (*e.g.*, *Zhang & Sievert, 2014*).

Pangenome analysis is most commonly applied to bacteria. Due to the explosion of data from metagenomes and microbiome studies, many bacterial taxa are well-sampled and can be associated with large sets of ecological or health-related metadata. Additionally, multiple software packages are available that facilitate automated inference of bacterial pangenomes, such as Anvi'o (*Eren et al., 2021*) and Roary (*Page et al., 2015*).

A typical pangenome analysis pipeline starts with two main steps: gene prediction and gene clustering. Often, workflows also include subsequent steps for function prediction, sequence alignment, and core gene identification. The accuracy of the two primary steps of inferring a pangenome is paramount. If a gene caller ignores an open reading frame (ORF) or inaccurately returns the end position of the ORF, genes may be truncated or merged. Errors in clustering—the process of placing related sequences into gene families—can include grouping unrelated genes or failing to place homologs in the same cluster. Together, these errors in gene calling and clustering may significantly impact identification of the "single-copy core genome" (SCG). The SCG is commonly used as the basis for phylogenetic inference, and excluding genes can mean missing important sequence variation and building less informative trees.

There is growing interest in applying pangenomic and phylogenomic workflows to bacteriophages (*e.g.*, *Edwards et al., 2019*; *Bellas et al., 2020*). Just as the deluge of metagenomic data has expanded bacterial comparative genomics, thousands of phage genomes are now published every year (*Roux et al., 2019*; *Dion, Oechslin & Moineau, 2020*). Because no single gene is conserved among all phage genomes, gene content profiles and gene sharing networks have become standard tools in virus taxonomy for identifying and comparing related viruses (*Bolduc et al., 2017*; *Shapiro & Putonti, 2018*). In the process, pangenomics has become an intrinsic component of phage bioinformatics.

Many of the potential sources of error for bacterial pangenome analysis are amplified when studying phages. First, phages are under-sampled despite regular publication of new genomes and identification of prophages within bacterial genomes (*Dion, Oechslin & Moineau, 2020*). Isolation, even of better-sampled groups through dedicated programs like SEA-PHAGES continues to discover novel viruses with genes lacking obvious homology to any known sequence (*Pope et al., 2015*). As a result, we often try to compare virus genomes that are more distantly related than expected for most pangenomic workflows. This can

make it difficult to recognize homologs between phage genomes that have low sequence identity. Further, many phages include intron-like sequences and homing endonucleases (*Belfort, 1990*; *Stoddard, 2005*). These selfish genetic elements interrupt genes and cause fragmented gene calls during annotation. Thus, the two main tasks of a pangenome analysis—gene identification and gene clustering—are more error-prone with phages than with bacteria.

Here, we describe a pipeline implemented in R, Rephine.r, for identifying and correcting common errors in the initial gene clusters and gene calls returned by pangenomic workflows. Given the results from a traditional pangenome analysis, Rephine.r: (1) merges gene clusters using Hidden Markov Models (HMMs) and (2) identifies fragmented gene calls to avoid the overprediction of paralogs and to improve sequence alignments. Each of the steps in Rephine.r can also be run separately for individual use cases that require only cluster merging or defragmentation. We demonstrate the value of Rephine.r using three phage taxa: the Tevenvirinae (*e.g.*, T4), the Studiervirinae (*e.g.*, T7), and the Pbunaviruses (*e.g.*, PB1). These virus groups represent a range of genome sizes and sampling depth, and each has at least 30 members with a RefSeq assembly. We show that correcting errors in gene cluster and gene fragmentation increases the size of the SCG in each case and enables inference of better-supported phylogenies. The tool is available through GitHub as a command line R script (https://www.github.com/coevoeco/Rephine.r) and includes utility scripts for returning the single-copy core genes and classifying the causes of gene fragmentation events.

## MATERIALS & METHODS

### Overview of the pipeline

The Rephine.r pipeline (summarized in Fig. 1) assumes the researcher has already completed a workflow for predicting gene clusters in a pangenome, such as the combination of blastp (*Altschul et al., 1990*) and MCL (*Enright, Van Dongen & Ouzounis, 2002*) implemented by Anvi'o (*Eren et al., 2021*) and other programs (*e.g.*, vConTACT *Bolduc et al., 2017* and Roary *Page et al., 2015*). In what follows, we use Anvi'o as the basis for initial pangenomes, as Anvi'o is both a popular tool for bacterial pangenomes and includes several useful commands for facilitating our corrections. Future updates will expand Rephine.r's compatibility with other tools.

Following initial gene clustering Rephine.r pipeline: (1) identifies and merges gene clusters containing distantly related homologs using HMMs, and (2) identifies fragmented gene calls that can be fused for the purpose of SCG inference and generating phylogenies. By default, Rephine.r will first run the cluster merging and defragmentation steps in tandem, produce a set of new clusters that combine the results of these corrections, and will then run a second round of defragmentation to identify any new cases that emerge due to the prior steps. Command line options are also offered for users that wish to run the HMM merging or fragment fusion steps individually. In addition to the main pipeline, we include two complementary scripts: getSCG.r returns the single-copy core genes and a concatenated alignment file for phylogenetics; fragclass.r categorizes the likely events that led to fragmented gene calls.
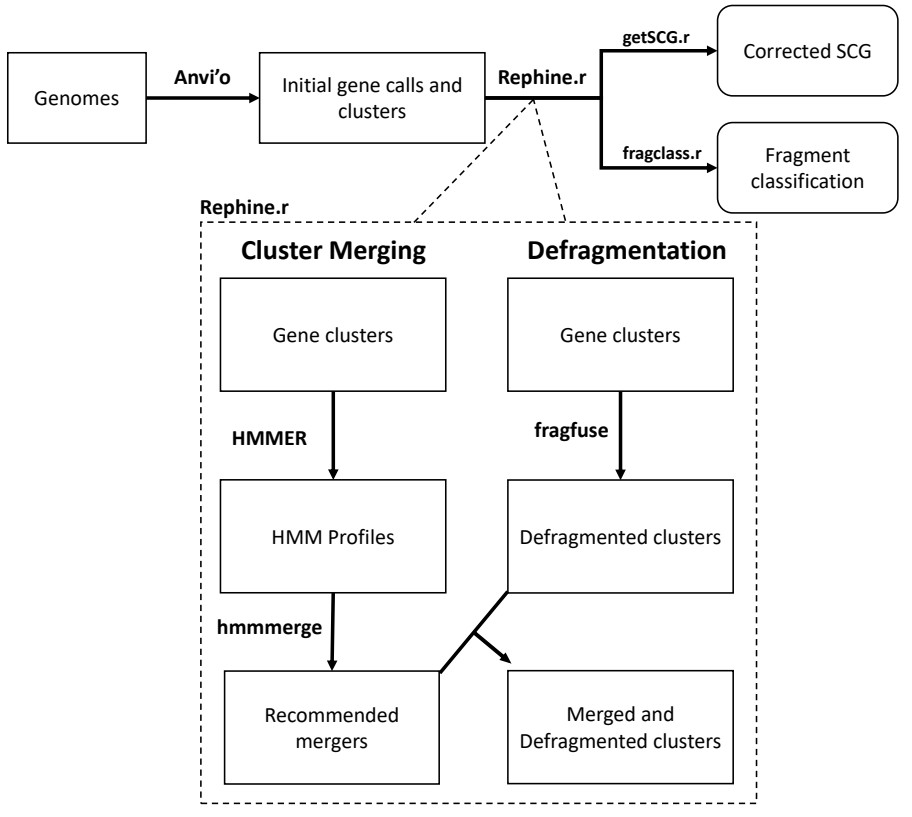

**Figure 1** **Flowchart of the Rephine.r pipeline.**

## Merging gene families with HMMs

Gene clustering based on sequence similarity relies on threshold criteria for defining when two sequences are related and for clustering related sequences into groups. In Anvi'o, the default identity heuristic is defined by the "minbit" score, the ratio of the BLAST bit score between two sequences and the minimum bit score from blasting each sequence against itself. This metric generally performs well, and for bacteria, where homologs are typically over 50% identical, it is especially successful. For phages, however, this approach can miss more distant homologs. Even using a 35% amino acid identity threshold (*Cresawn et al., 2011*; *Shapiro & Putonti, 2018*), we may miss cases that only appear related when viewing alignments or comparing phage genes by structure or synteny. Unfortunately, it is not as simple as specifying a lower minbit threshold, since doing so will also increase the number of unrelated genes that are clustered together erroneously.

Given the initial gene clusters returned by Anvi'o, Rephine.r builds separate HMM profiles for each cluster using the hmmbuild function from HMMER (*Eddy, 1998*) and converts the concatenated HMM profiles into a database with hmmpress. The script then uses hmmscan to compare every original gene call against each HMM profile. This step is expected to be more sensitive for recognizing distant homologs than the initial blastp, as the HMM profiles make use of variation from multiple members of the same cluster.

Significant hits are then defined as follows: for each original gene cluster, the "minimum self-bit" (or "selfbit") score is recorded as the minimum of the bit scores for each of the gene calls that was initially assigned to that cluster by MCL. This selfbit score then serves as a profile-specific significance threshold. Any gene call that was originally assigned to another cluster but has a bit score greater than this value is then used to establish a putative connection between gene clusters. We also include the option of specifying an absolute minimum bit score as an additional criterion. These connections are recorded in the form of a network edgelist linking gene calls to gene clusters. Next, this edgelist is relabeled to define edges between the original gene clusters that share putative homologs. Finally, this edgelist is used to generate a network with the R (*R Core Team, 2013*) package igraph (*Csardi et al. , 2006*), and the connected components are returned with the function "components". The result defines sets of the original gene clusters that are suitable for merging into a single, larger cluster.

### Identifying fragmented gene calls

To find fragmented genes, Rephine.r first identifies every gene cluster that includes at least two sequences from the same genome. These sequences may represent true duplicates or paralogs, or they may be separate pieces of the same original sequence that have been split by one of several processes, including: a frameshift due to an indel, insertion of a selfish genetic element, or being artificially split across the ends of the genome when it was reported to GenBank. This third case may also arise as an artifact of the two other mechanisms. For any of these scenarios, the two pieces of the gene will be notable in two ways: (1) they will align with separate parts of the gene in a multiple sequence alignment, with one piece corresponding to an N-terminal fragment, and the other to the C-terminus; (2) they should have lower sequence similarity to each other than to the average comparison with other sequences in the multiple sequence alignment. Fig. 2A illustrates how a fragmented gene may appear in an alignment.

Given clusters with potential fragments, every gene call within an affected cluster is compared using blastp to every other gene call in the same cluster. For the two focal gene calls from a potential fragmented gene, the bit score from their blast alignment is compared to the mean bit score for other blast results within the gene cluster. We defined the ratio of this pairwise blast to the cluster average as the "relative bit" (or "relbit"). Mathematically, for potential fragments A and B within a gene cluster G, this is defined as:

$$relbit\,(A,B) = \frac{bit\,(A,B)}{\overline{bit}\,(A,G)}, relbit\,(B,A) = \frac{bit\,(B,A)}{\overline{bit}\,(B,G)} \tag{1}$$

where the overbar refers to the mean. The maximum of these relbit values is then used as a criterion for judging similarity between A and B. If this value is below a chosen threshold, the ORFs are considered to be sufficiently dissimilar.

Rephine.r also compares the extent of overlap within the pairwise alignment space between each potential paralog. This step is needed, because dissimilar gene fragments may still have overlaps in the alignment due to alignment errors or if the original fragmentation event was caused by a short duplication. To quantify this overlap, the "percent overlap" is
_______________

**A**
```
>NC_007810_79
MTAKYYSPDDLVTPQEFADPQFAAINQKRFDLYIDLRVQGYSSWRVFRAIWGEEHMDGPA
QARIFAMESNPYYRKQFKAKLNATRTSDLWNPKTALHELLQMVRDPTVKDSSRLSAIKEL
NVLAEITFVDESGKTRVGRGLADFYASEAEAQTATVAAAAEANGYVQDGEEGDFPSPTPE
PTEEDRANPIQT
>NC_041902_47
-MTKFYSPDDLVTPQEFADPHFAAINQKRFDLYIDLRVQGYSSWRVFRAIWGEEHMDGPA
QARIFAMESNPYYRKQFKAKLNATKR-----------------PICGIQRRRST----
------------------NSSKWFVTP------------------------PSRTPA
VCRPSRN-----
>NC_041902_46
------------------------------------------------------------
--------------------------------------------MVRDPTVKDSSRLSAIKEL
NVLAEITFVDESGKTRIGRGLADFYASEAEAQTATVAAAAEANSYVPEGEEGDFPSPTPE
PTEEDRANPI--
```

**B**
```
>NC_007810_79
MTAKYYSPDDLVTPQEFADPQFAAINQKRFDLYIDLRVQGYSSWRVFRAIWGEEHMDGPA
QARIFAMESNPYYRKQFKAKLNATR------------TSDLWN-------PKTALHELLQ
MVRDPTVKDSSRLSAIKELNVLAEITFVDESGKTRVGRGLADFYASEAEAQTATVAAAAE
ANGYVQDGEEGDFPSPTPEPTEEDRANPIQT
>NC_041902_47:46
-MTKFYSPDDLVTPQEFADPHFAAINQKRFDLYIDLRVQGYSSWRVFRAIWGEEHMDGPA
QARIFAMESNPYYRKQFKAKLNATKRPICGIQRRRSTNSSKWFVTPPSRTPAVCRPSRNX
MVRDPTVKDSSRLSAIKELNVLAEITFVDESGKTRIGRGLADFYASEAEAQTATVAAAAE
ANSYVPEGEEGDFPSPTPEPTEEDRANPI--
```

**Figure 2  Fragmented gene calls can be identified from alignments.** (A) An original multiple sequence alignment where the gene from NC_041902 has been split into two fragments by an indel. (B) The corrected alignment following Rephine.r. Highlighted colors are used to indicate regions of each fragment and where they correspond within an intact homolog.

calculated as the size of the ORFs' intersection within the alignment divided by the number of unique, aligned positions between the two sequences. In mathematical terms, for a gene with potential fragments A and B, we define:

$$PercentOverlap = \frac{|A \cap B|}{|A \cup B|} \tag{2}$$

where the size terms are based solely on the aligned positions within the multiple sequence alignment.

Ultimately, sequence pairs with low relative bit scores ("relbit") and low percent overlaps ("percoverlap") are the likeliest to fit our expectations of a fragmented gene call. In practice, we implemented default parameters for these criteria of 0.25 for "relbit" and 0.25 for "percoverlap." These choices are based on plotting values of each parameter (Fig. S1) from the test cases described below and identifying a set of points that weakly cluster together in the graph. When checked manually, each of these genes appeared to correspond to fragmented calls, whereas nearby points in the graph included potential

errors. These parameters can be adjusted at the command line, and we would encourage others to visually inspect their alignments.

Once fragmented genes are identified, a new FASTA file is created in which the original pieces of the full-length gene are artificially spliced (or "fused") into a single gene call. To preserve the original event that separated the sequences, the script inserts an "X" between the two pieces of the gene. New alignments are then made with MUSCLE (*Edgar, 2004*) for each affected gene cluster, with these X's imposing a gap in the alignment (see Fig. 2B for an illustration of this step). If desired, the user can then use the additional script, getSCG.r, to return a list of the single-copy core gene clusters, along with a concatenated alignment file that is suitable for phylogenetics. The script, fragclass.r, can also be used to obtain a table summarizing predicted causes for each type of fragment based on the separation between the original gene calls.

## Virus genomic data

Phages in the subfamily Studiervirinae (family Autographiviridae), the subfamily Tevenvirinae (family Myoviridae), and the genus Pbunavirus (family Myoviridae) were chosen as well-studied examples for testing Rephine.r. We downloaded all available RefSeq genomes from each of these taxa from the National Center for Biotechnology Information's (NCBI) genome browser (as of February 2021). This data set included 145 Studierviruses, 127 Tevenviruses, and 38 Pbunaviruses (a full list of accessions is included in Table S1). The Studiervirinae (*e.g.*, phages T3 and T7) and the Tevenvirinae (*e.g.*, phage T4) are among the best-studied phage subfamilies and include characterized examples of introns and homing endonucleases (*Chu et al., 1986*; *Belle, Landthaler & Shub, 2002*; *Bonocora & Shub, 2004*; *Petrov, Ratnayaka & Karam, 2010*). These features made these two subfamilies ideal for testing methods for identifying distant homologs and fragmented gene calls. The Pbunaviruses were chosen due to the relatively large number of available genomes at the genus level, offering a less diverse contrast to the other phage groups.

## Initial pangenome workflow with Anvi'o

We built an initial pangenome for each phage group using Anvi'o v6.2 (*Eren et al., 2021*) following the standard pangenomics workflow (https://merenlab.org/2016/11/08/pangenomics-v2/), which uses Prodigal (*Hyatt et al., 2010*) for gene calling. Alternative gene callers can also be used, and these gene calls can be imported into Anvi'o as part of the anvi-gen-contigs-database program. The "–use-ncbi-blast" flag was specified for the anvi-pan-genome command. Due to the large genetic diversity of phages, we set the minbit threshold to 0.35, based on prior work (*Cresawn et al., 2011*; *Shapiro & Putonti, 2018*).

## Phylogenetics

Maximum likelihood phylogenies were estimated using IQTREE v2.0.3 (*Nguyen et al., 2015*) with ModelFinder (*Kalyaanamoorthy et al., 2017*) to automate choosing the optimal substitution model for each tree. For each of the three virus groups, trees were built based on concatenated alignments for the original SCGs and again following Rephine.r using the expanded SCGs. Tree summary statistics were computed in R using the ape package (*Paradis, Claude & Strimmer, 2004*) and drawn using ggtree (*Yu et al., 2017*).
| Table 1 Summary of results of running Rephine.r for each phage group. | | | |
|---|---|---|---|
| | **Studiervirinae** | **Tevenvirinae** | **Pbunaviruses** |
| Number of genomes | 145 | 127 | 30 |
| Mean genome size | 39696 | 174775 | 66068 |
| Initial gene calls | 6956 | 35436 | 3540 |
| Initial gene clusters | 558 | 4067 | 195 |
| Initial core genes | 12 | 27 | 28 |
| Initial SCG size | 3 | 13 | 19 |
| New clusters after merging | 16 | 64 | 2 |
| Clusters involved in a merger | 63 | 270 | 5 |
| Biggest merger | 7 | 30 | 3 |
| Core genes after merging | 14 | 37 | 28 |
| SCG size after merging | 3 | 13 | 19 |
| Defragmented clusters | 14 | 99 | 17 |
| SCG size after fusion and merge | 8 | 22 | 26 |
| Additional fusions after merge | 1 | 7 | 1 |
| New core genes after final fusion | 0 | 0 | 0 |
| Total SCG gain | 5 | 9 | 7 |
| Mean tree support before | 77.14 | 87.24 | 63.6 |
| Mean tree support after | 90.55 | 93.44 | 69.57 |

## Code Availability

All code for this work is provided on GitHub (https://github.com/coevoeco/Rephine.r).The code includes a walkthrough for running Rephine.r following a standard Anvi'o workflow, as well as utility scripts, getSCG.r and fragclass.r, that provide additional output of the SCG genes and predicted causes of fragmentation events.

## RESULTS

To test the Rephine.r pipeline, we downloaded all available RefSeq genomes for the Studiervirinae, Tevenvirinae, and Pbunaviruses from NCBI. We then followed the standard pangenomic workflow for Anvi'o to facilitate initial MCL clustering based on blastp scores. Results and basic information about these taxa are summarized in Table 1. Across all Studieviruses, there were only 12 core genes, of which three were single-copy. Tevenviruses included 27 core genes (13 single-copy), and the Pbunaviruses had 28 core genes (19 single-copy).

We ran Rephine.r with default settings, which first predicts fragmented gene calls within each gene cluster. In tandem, it identifies related gene clusters using HMMs. It then combines the results from these steps to produce new merged clusters with corrections for fragmented genes. Last, it runs a second defragmentation step to identify instances where fragmented gene calls were originally split into separate gene clusters. We examined results to see how the core genome changed after each step and how the final SCG affected phylogenetic inference.
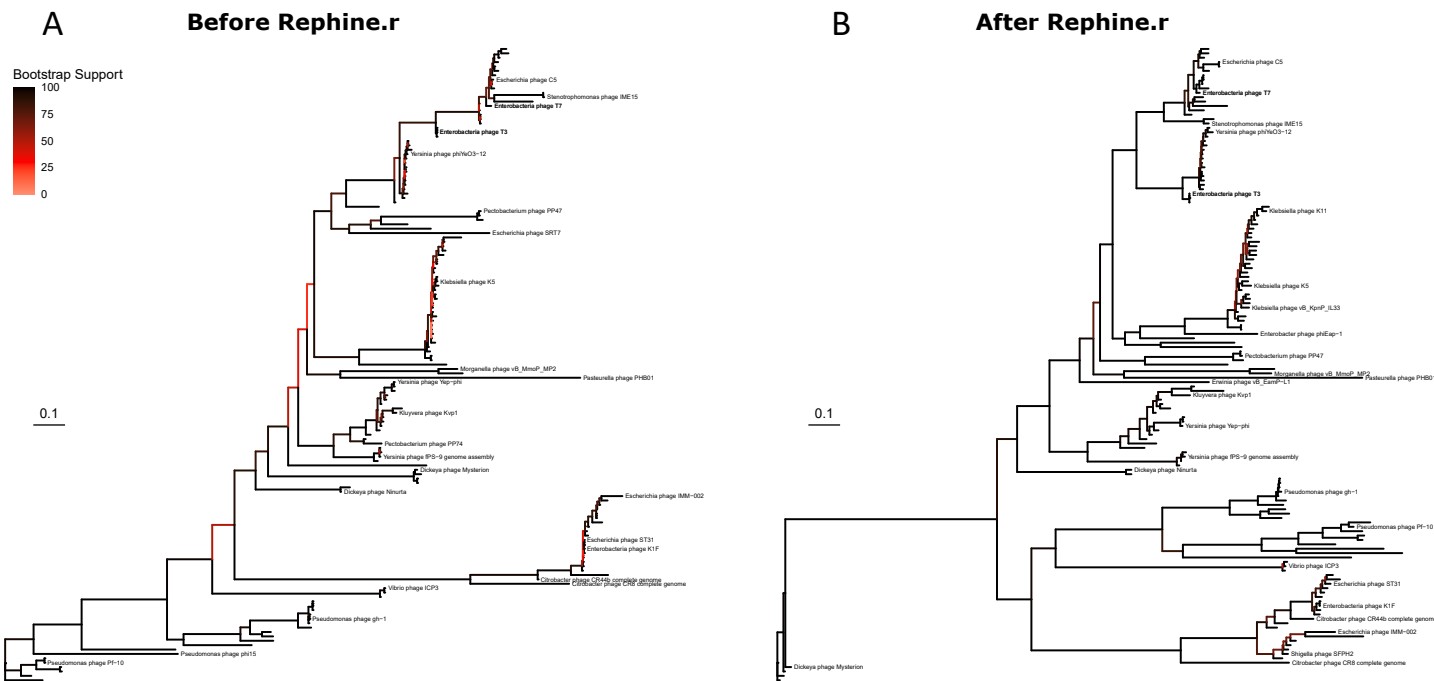

**Figure 3** **Studiervirinae phylogeny before (A) and after (B) using Rephine.r to correct the SCG.** Bootstrap support is shown by coloring branches preceding nodes, with low support (from 0 to 70) ranging from white to red. Increasing the size of the SCG reduced the number of low-support branches.

The initial HMM merging step resulted in two additional core genes for Studierviruses and 10 additional core genes for Tevenviruses but no new single-copy core genes for any of the virus groups. Notably, several mergers involved more than two gene clusters. In one case for the Tevenvirinae, 30 separate gene clusters were merged, corresponding to the phage tail fiber. Defragmenting gene calls expanded the SCG for each taxon, increasing the Studiervirinae SCG to 8 genes, the Tevenvirinae to 22 genes, and the Pbunaviruses to 26 genes (all but two of the Pbunavirus core genes). The final round of defragmentation identified additional fragmented genes but no additional core genes.

We then built phylogenies for each taxon with the original SCGs and with expanded SCGs following Rephine.r. With only three single-copy core genes, the initial Studiervirinae tree contained multiple unresolved polytomies and branches with poor support (Fig. 3A). The updated tree based on eight genes had improved overall bootstrap support and displayed greater resolution of closely related genomes (Fig. 3B). (A version of the tree with all labels is provided in Fig. S2). Trees for the Tevenvirinae and Pbunaviruses (Figs. S3 and S4) also had improved bootstrap support. In the case of the Pbunaviruses, the tree remains poorly resolved with very short branches, despite being built from the most genes, as there was insufficient variation among the viruses from this genus.

Last, we checked the results from gene call defragmentation for known instances of introns and homing endonucleases in the Studiervirinae and Tevenvirinae. These include interruptions to DNA polymerase in members of Studiervirinae (*Bonocora & Shub, 2004*)

and Tevenvirinae (*Petrov, Ratnayaka & Karam, 2010*) and thymidylate synthase in T4 (*Chu et al., 1986*). After running Rephine.r, we identified a single-copy core gene that corresponded to each gene of interest. In each case, inclusion in the SCG was only possible after fragment identification.

## DISCUSSION

We describe Rephine.r, a pipeline for improving results of phage pangenome analysis by merging gene clusters containing distant homologs and correcting gene calls that have been fragmented or interrupted by selfish genetic elements. Using the Tevenvirinae, Studiervirinae, and Pbunaviruses as test cases, we show how this process expands the putative SCG for each group, enabling more accurate estimates of gene conservation. For the Tevenvirinae and Studiervirinae, this also improved the quality of the phylogenies, whereas for Pbunaviruses there was still insufficient variation among the genomes to produce a reliable tree.

The present work provides a first step for expanding the usage of phylogenetics with diverse phage genomes. A key concept that we include (which we took advantage of using manual corrections previously (*Shapiro & Putonti, 2020*) is the use of artificially spliced sequences following the identification of interrupted genes. This type of correction is unsurprising when working with eukaryotic exons, but it is generally ignored with microbes, because we often fail to appreciate that intron-like sequences are common features of many phages. Biologically, it is uncertain how often these interrupted genes remain functional or if the separated ORFs correspond to separate functions. However, several studies report fully functional, single protein products for phage genes separated by introns (*Belfort, 1990*) or inteins (*Kelley et al., 2016*), as well as at least one case where a gene split by a homing endonuclease remains active (*Friedrich et al., 2007*). Though these ORFs may be interrupted by over 1,000 nucleotides, these interruptions likely correspond to a single mutational event, and the ORFs should still be treated as a single gene when reconstructing the SCG and an associated phylogeny. In both the Studiervirinae and the Tevenvirinae, our approach accurately recognized known homing endonucleases and introns, and these genes remain the most common multi-copy core genes following Rephine.r. How interrupted genes are interpreted in functional genomics studies is an important question, and these fragmented genes should be treated with additional care when reporting the functional repertoire of genomes.

It is important to note that we have focused our application of Rephine.r on test cases involving single-contig, RefSeq assemblies. In the case of draft genome assemblies comprised of multiple contigs (less common for phages under 100 kb), we expect to observe instances where a gene call is separated into different ORFs on different contigs. These errors will result in overestimating gene content and incorrect predictions of paralogous sequences. Similar issues have been noted to cause errors in the analysis of gene content evolution in eukaryotes (*Denton et al., 2014*). The current implementation of gene defragmentation in Rephine.r should successfully resolve many of these mistakes, and it may offer a future approach for consolidating contigs in assemblies. For instance, suppose

a gene is split by a transposase that includes short palindromic repeats. These regions are difficult to assemble with short reads and may lead to one contig ending with half of the original gene, while a second contig starts with the transposase and the remainder of the gene. Scaffolding these contigs can be challenging, but by recognizing gene fragments, it may be possible to resolve the assembly.

Last, bacterial pangenome workflows typically do not account for specific issues that may arise for prophage regions, such as errors in clustering and gene fragmentation that we observe in the genomes of phage isolates. Our expectation is that these same errors will impact prophages, and future work will need to consider how these issues may impact the accuracy of bacterial pangenomes. Moreover, bacterial genes themselves can be interrupted by mobile genetic elements (including phages), and Rephine.r should offer a novel approach for identifying these events.

## CONCLUSIONS

The Rephine.r pipeline offers an efficient means to identify and correct errors in phage pangenomes caused by incomplete gene clustering and fragmented gene calls. Correcting these errors, in particular for cases of genes interrupted by selfish genetic elements, increases the size of the SCG in each of our test cases. These corrections provide more genetic variation for improved phylogenetic inference and are especially useful for large, diverse phage groups where standard methods produce limited core genomes and poorly resolved phylogenies.

## ACKNOWLEDGEMENTS

We are grateful to the members of the Putonti Lab for feedback on this work.

### Funding
This work was supported by the National Science Foundation (1661357 to Catherine Putonti). The funders had no role in study design, data collection and analysis, decision to publish, or preparation of the manuscript.

### Grant Disclosures
The following grant information was disclosed by the authors:
National Science Foundation: 1661357.

### Competing Interests
The authors declare there are no competing interests.

### Author Contributions
- Jason W Shapiro conceived and designed the experiments, performed the experiments, analyzed the data, prepared figures and/or tables, authored or reviewed drafts of the paper, and approved the final draft.

- Catherine Putonti conceived and designed the experiments, authored or reviewed drafts of the paper, and approved the final draft.

## Data Availability

All raw data is available at figshare: https://doi.org/10.6084/m9.figshare.14485389.v1

The code is available at GitHub: https://github.com/coevoeco/Rephine.r.

## Supplemental Information

Supplemental information for this article can be found online at http://dx.doi.org/10.7717/peerj.11950#supplemental-information.

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
