# Peer review of "Rephine.r: a pipeline for correcting gene calls and clusters to improve phage pangenomes and phylogenies"

_PeerJ, doi:10.7717/peerj.11950_

## Round 0.1 · original submission · Major Revisions

· Academic Editor

Major Revisions

Dear Drs. Shapiro and Putonti:

Thanks for submitting your manuscript to PeerJ. I have now received two independent reviews of your work, and as you will see, the reviewers raised some relatively minor concerns about the research. This is great and indicates optimism for your work and the potential impact it will have on research studying phage gene annotation.

While the concerns of the reviewers are relatively minor, this is a major revision to ensure that the original reviewers have a chance to evaluate your responses to their concerns. There are many suggestions, which I am sure will greatly improve your manuscript once addressed.

Therefore, I am recommending that you revise your manuscript, accordingly, taking into account all of the issues raised by the reviewers. I do believe that your manuscript will be greatly improved once these issues are addressed.

Good luck with your revision,

-joe

Reviewer 1 ·

Basic reporting

no comment

Experimental design

no comment

Validity of the findings

no comment

Additional comments

This report describes Rephine.r, a pipeline to identify split genes in phage genomes, fuse them back and use them for the determination of core genes among phage clades. The tool is then applied to a phage genus Pbunavirus, and two subfamilies, Tevenvirinae and Studiervirinae. The report finally shows the phylogeny improvements provided by the increased amount of single core genes after such cleaning up.
This work provides a very timely and useful tool for phage biology as a whole that will also help in the field of phage taxonomy. The methodology is clean and well exposed, I have no major point to raise, only suggestions for improvements.
1. A correlative problem to gene split in phage genome analyses is the gene calling method itself, which needs to be adapted for phages. This is not addressed in this work, as it takes as entry point the output of the Anvi’o pipeline. Please describe in the Methods how is Anvi’o performing this step (which program, what minimal orf size, is some level of orf overlap permitted). Provide a comparison of the number of orfs called by Anvi’o and those present in the Genbank files of reference T4, PB1 and T7 genomes, which have been studied in depth. Is there an option for alternative codes (as some huge phages appear to use such codes)?
2. Lines 199-211: some clarification is needed. Line 199, « the extent of overlap within the alignment space », is this the multiple alignment? Line 204 : « within the alignment », this is again the multiple alignment probably. In addition, the explanation of this overlap calculation is somewhat unclear. It seems one is dividing the size of ORF intersection by the size… of ORF overlap (so it is the same)? Should A U B not be the size of the aligned positions in A and B?
3. Table 1 and Figure 1: could you use the same term of « defragmented clusters » in both cases (Table 1 uses instead « gene clusters with a fusion »), or is there a subtle nuance here?
4. Give the final SGC list (with annotations) for each clade (and indicate the new ones). Such information is insightful for taxonomists.
5. Fig3 and Suppl Fig2: label the leaves, at least one phage in each group. Are the Tevenvirinae shown at the very root of the tree real ones, or rather falsely classified phages?
6. The number of remaining paralogs in each clade is high. What kind of function do they encode?
7. Check the supplementary figure legends, I could not find them but they may be hidden somewhere in the files.

·

Basic reporting

This manuscript introduces Rephine.r, which is a novel type of tool to assist phage pangenome analysis. Pangenome analysis of phages are not very popular so far but must be promising to characterize phage lineages. The manuscript is clearly written, and the developed pipeline is well designed to overcome the potential difficulty of phage pangenome analysis.

Experimental design

The main pipeline of Rephine.r is composed of two parts, (1) merges remotely related gene clusters and (2) fuses fragmented genes. Each component is effective for phage pangenome analysis. Especially, the second component includes high originality, and it is applicable for various cases as is discussed. However, I considered that only ‘the degree of overlap in the alignment’ is enough to discriminate the true and false gene separation and did not understand why sequence dissimilarity is needed to be evaluated. It would be better to discuss the case of low overlap and high similarity (more than 10 cases exist, as shown in Suppl. Fig 1).

Validity of the findings

All conclusions are valid.

Additional comments

1. This work provides much improvement to phage pangenome analysis. Nevertheless, I suggest one point that might be better to be discussed. If a gene is completely missed in the original gene call, the gene is not rescued by the current implementation of Rephine.r. Merged use of multiple gene caller, including phage specific ones such as PHANOTATE, will provide further improvement. Is it possible to incorporate this step into Rephine.r?
2. L169 includes an empty parentheses.

---

## Round 0.2 · accepted · Accept

· Academic Editor

Accept

Dear Drs. Shapiro and Putonti:

Thanks for revising your manuscript based on the concerns raised by the reviewers. I now believe that your manuscript is suitable for publication. Congratulations! I look forward to seeing this work in print, and I anticipate it being an important resource for groups studying phage gene annotation. Thanks again for choosing PeerJ to publish such important work.

Best,

-joe

Reviewer 1 ·

Basic reporting

The manuscript is now ready for publication.

Experimental design

OK

Validity of the findings

OK

·

Basic reporting

no comment

Experimental design

no comment

Validity of the findings

no comment

Additional comments

The manuscript is significantly improved.